# Parameter elimination in particle Gibbs sampling

**Anna Wigren**
Department of Information Technology
Uppsala University, Sweden
`anna.wigren@it.uu.se`

**Riccardo Sven Risuleo**
Department of Information Technology
Uppsala University, Sweden
`riccardo.risuleo@it.uu.se`

**Lawrence Murray**
Uber AI
San Francisco, CA, USA
`lawrence.murray@uber.com`

**Fredrik Lindsten**
Division of Statistics and Machine Learning
Linköping University, Sweden
`fredrik.lindsten@liu.se`

## Abstract

Bayesian inference in state-space models is challenging due to high-dimensional state trajectories. A viable approach is particle Markov chain Monte Carlo, combining MCMC and sequential Monte Carlo to form "exact approximations" to otherwise intractable MCMC methods. The performance of the approximation is limited to that of the exact method. We focus on particle Gibbs and particle Gibbs with ancestor sampling, improving their performance beyond that of the underlying Gibbs sampler (which they approximate) by marginalizing out one or more parameters. This is possible when the parameter prior is conjugate to the complete data likelihood. Marginalization yields a non-Markovian model for inference, but we show that, in contrast to the general case, this method still scales linearly in time. While marginalization can be cumbersome to implement, recent advances in probabilistic programming have enabled its automation. We demonstrate how the marginalized methods are viable as efficient inference backends in probabilistic programming, and demonstrate with examples in ecology and epidemiology.

## 1 Introduction

State-space models (SSMs) are a well-studied topic with applications in climatology [3], robotics [8], ecology [29], and epidemiology [31], to mention just a few. In this paper we propose a new method for performing Bayesian inference in such models. In SSMs, a latent (hidden) state process $x_t$ is observed through a second process $y_t$. The state process is assigned an initial density $x_0 \sim p(x_0)$, and evolves in time according to a transition density $x_t \sim p(x_t|x_{t-1}, \theta)$, where $\theta$ are parameters with prior density $p(\theta)$. Given the latent states $x_t$, the observations are assumed independent with density $p(y_t|x_t, \theta)$. We wish to infer the joint posterior, $p(x_{0:T}, \theta|y_{1:T})$, for the states $x_{0:T}$ and the parameters $\theta$, given a set of observations $y_{1:T} = \{y_1, \dots, y_T\}$. Unfortunately, computing this posterior distribution exactly is not analytically tractable for general non-linear, non-Gaussian models, so we must resort to approximations.

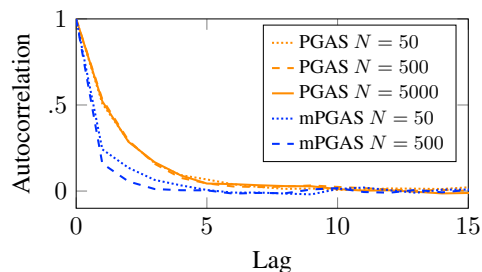

Figure 1: The autocorrelation function (ACF) for standard PGAS converges to that of the hypothetical Gibbs sampler as $N \rightarrow \infty$, whereas mPGAS will produce iid draws in the limit, i.e., the ACF will drop to zero at lag one for large $N$. Similar results hold for PG and mPG, see Supplementary E.

Markov chain Monte Carlo (MCMC) [e.g. 32] is a popular choice for Bayesian inference. The motivation behind our new method is based on one such MCMC method: the Gibbs sampler. In the Gibbs sampler, samples from the posterior $p(x_{0:T}, \theta | y_{1:T})$ are generated by alternating between sampling the states from $x'_{0:T} \sim p(x_{0:T} | y_{1:T}, \theta')$, and the parameters from $\theta' \sim p(\theta | x'_{0:T}, y_{1:T})$. Sampling the parameters is often manageable, but sampling the states is challenging, owing to the distribution being high-dimensional. A possible remedy is to use particle Markov chain Monte Carlo (PMCMC) methods [2], in which sequential Monte Carlo (SMC) is used to approximate sampling from the high-dimensional distribution. Particle Gibbs (PG) [2] is a PMCMC algorithm that mimics the Gibbs sampler. Efficient extensions, such as particle Gibbs with ancestor sampling (PGAS), have also been proposed, reducing the computational cost from quadratic to linear in the number of timesteps, $T$, in favorable conditions [18, 20].

PG and PGAS have proven to be efficient in many challenging situations [e.g. 19, 22, 39, 40]. Nevertheless, being "exact approximations" [1] of the (possibly intractable) Gibbs sampler, they can never outperform it. In essence, this means that when the number of particles used in their SMC component approaches infinity, PG and PGAS will approach the hypothetical Gibbs sampler in terms of autocorrelation, but can never surpass it. This is illustrated in Figure 1, orange curve (for details on the model, see Section 3.2). Ideally, independent samples from the target distribution are desired, but the often strong dependence between the parameters $\theta$ and the states $x_{0:T}$ in the hypothetical Gibbs sampler leads to correlated samples also for PG and PGAS.

In marginalized Gibbs sampling we propose to marginalize out the parameters in the state update, ideally alternating between sampling the states $x'_{0:T} \sim p(x_{0:T} | y_{1:T})$ and sampling the parameters $\theta' \sim p(\theta | x'_{0:T}, y_{1:T})$ (note that an alternative is to sample only the state trajectories $\{x^i_{0:T}\}^M_{i=1}$, where $M$ is the number of MCMC steps, and then estimate the posterior of $\theta$ as a mixture of densities, where each component is $p(\theta | x^i_{0:T}, y_{0:T})$). The state update is thus independent of the parameters and this hypothetical marginalized Gibbs sampler will effectively generate independent samples from the target distribution. However, like for the unmarginalized hypothetical Gibbs sampler, the distribution for sampling the states is not available in closed form. To address this issue, we derive marginalized versions of PG and PGAS (hereon referred to as mPG/mPGAS). Analogous to the unmarginalized case, with an increasing number of particles, mPG and mPGAS will approach the hypothetical marginalized Gibbs sampler – that is, a sampler generating independent samples from the target. This behavior is illustrated in Figure 1, blue curve.

Marginalization is possible if the SSM has a conjugacy relation between the parameter prior and the complete data likelihood, that is, the conditional $p(\theta | x_{0:T}, y_{1:T})$ has the same functional form as the prior $p(\theta)$. However, even for such models there is a price to pay for marginalization: it turns the Markovian dependencies, central to the SSM when conditioned on the parameters, into non-Markovian dependencies for both states and observations. This will make it harder to apply conventional MCMC methods, whereas PMCMC methods have proven to be better suited for models of this type [20]. In Section 3 we derive the algorithmic expressions for mPG and mPGAS for this family of models. The necessary updates in each step in the marginalized SMC algorithm can be done using sufficient statistics, which enables the computation time of mPG and mPGAS to scale linearly with the number of observations, despite the non-Markovian dependencies. The class of conjugate SSMs includes many common models, but is still somewhat restrictive. In Section 4, we discuss some extensions to make the framework more generally applicable and provide numerical illustrations.

Marginalization of static parameters in the context of SMC has been studied by [5, 36] for the purpose of online Bayesian parameter learning. To what extent these methods suffer from the well-known path degeneracy issue of SMC has been a topic of debate, see e.g. [7]. Since our proposed method is based on PMCMC, and in particular PGAS, it is more robust to path degeneracy, see [20]. The Rao-Blackwellized particle filter [6, 9] also makes use of marginalization, but for marginalizing part of the state vector using conditional Kalman filters.

In practice, deriving the conjugacy relations can be quite involved. However, recent developments in probabilistic programming have enabled automatic marginalization [see e.g. 16, 26, 28], which significantly improves the usability of our proposed method. Probabilistic programming considers the way in which probabilistic models and inference algorithms may be expressed in universal programming languages, formally extending the expressive power of graphical models. There are by now quite a number of probabilistic programming languages. Examples that can support SMC-based

methods, such as those considered here, include LibBi [23], BiiPS [37], Venture [21], Anglican [38], WebPPL [14], Figaro [30], Turing [13], and Birch [24]. A language can implement PG/PGAS combined with automatic marginalization to realize our proposed method. We have implemented PG, mPG, PGAS and mPGAS in Birch [24] and provide examples to illustrate their efficiency in Section 4.2 and 4.3.

## 2 Background on SMC

In PG and PGAS, the state update is approximated using SMC, therefore we provide a brief summary of the SMC algorithm before introducing the proposed method. For a more extensive introduction, see e.g. [4, 15]. Consider a sequence of probability densities $\bar{\gamma}_{\theta,t}(x_{0:t})$ expressed as

$$\bar{\gamma}_{\theta,t}(x_{0:t}) = \frac{\gamma_{\theta,t}(x_{0:t})}{Z_{\theta,t}}, \qquad t = 1, 2, \ldots \tag{1}$$

where $\gamma_{\theta,t}$ are the corresponding unnormalized densities, which we assume can be evaluated pointwise, and $Z_{\theta,t}$ is a normalizing constant. For a SSM, the target density of interest is often $p(x_{0:t}|y_{1:t}, \theta)$, which implies $\gamma_{\theta,t} = p(x_{0:t}, y_{1:t}|\theta)$ and $Z_{\theta,t} = p(y_{1:t}|\theta)$. SMC methods approximate the target density (1) using a set of $N$ weighted samples (or particles) $\{x_{0:t}^i, \bar{w}_t^i\}_{i=1}^N$, generated according to Algorithm 1. When moving to the next distribution in the sequence, all particles are resampled by choosing an ancestor trajectory $x_{0:t-1}^{a_t^i}$ from the previous step in time according to the respective weights $\bar{w}_{t-1}^i$ of the possible ancestors. SMC is based on importance sampling and the resampled particles are therefore propagated to the next time step using a proposal distribution, $q_{\theta,t}(x_t|x_{0:t-1})$, chosen by the user. A common choice for SSMs is to use the bootstrap proposal, which equates to propagating according to the transition density $p(x_t|x_{t-1}, \theta)$, but other more refined choices, such as the optimal proposal (see e.g. [10]), are also possible. Finally, the (unnormalized) importance weights for the propagated particles are computed using the weight function

$$\omega_{\theta,t}(x_{0:t}) = \frac{\gamma_{\theta,t}(x_{0:t})}{\gamma_{\theta,t-1}(x_{0:t-1})q_{\theta,t}(x_t|x_{0:t-1})}. \tag{2}$$

---

**Algorithm 1** SMC (all steps for $i = 1, \ldots, N$)

---

1: *Initialize:* Draw $x_0^i \sim q_0(x_0)$, set $w_0^i = \gamma_{\theta,0}(x_0^i)/q_0(x_0^i)$, normalize $\bar{w}_0^i = w_0^i / \sum_{j=1}^N w_0^j$
2: **for** $t = 1 \ldots T$ **do**
3:     *Resample:* Draw $a_t^i \sim \mathcal{C}(\{\bar{w}_{t-1}^i\}_{i=1}^N)$, where $\mathcal{C}$ is the categorical distribution.
4:     *Propagate:* Simulate $x_t^i \sim q_{\theta,t}(x_t|x_{0:t-1}^{a_t^i})$.
5:     *Update:* Set $w_t^i = \omega_{\theta,t}(x_{0:t}^i)$ according to (2) and normalize $\bar{w}_t^i = w_t^i / \sum_{j=1}^N w_t^j$
6: **end for**

---

## 3 Method

In this section, we first specify the class of models we consider, and then we show how to marginalize the SMC algorithm and derive mPG and mPGAS for this class of models.

### 3.1 Conjugate models and marginalized SMC

The SMC framework presented in Section 2 is in a general form and can be directly applied to the marginalized state update by defining the unnormalized target distribution as $\gamma_t(x_{0:t}) = p(x_{0:t}, y_{1:t})$ in (1) and then applying Algorithm 1. The computation of the importance weights (step 5 in Algorithm 1), however, turns out to be problematic in marginalized SSMs. To see why, note that the unnormalized target density can be factorized into $p(x_{0:t}, y_{1:t}) = p(x_0) \prod_{k=1}^t p(x_k, y_k|x_{0:k-1}, y_{1:k-1})$. The weights (2) become

$$\omega_t(x_{0:t}) = \frac{p(x_t, y_t|x_{0:t-1}, y_{1:t-1})}{q_t(x_t|x_{0:t-1})} \tag{3}$$

where the numerator (and possibly also the denominator depending on the choice of proposal) is non-Markovian. The marginal joint density of states and observations can be written

$$p(x_t, y_t | x_{0:t-1}, y_{1:t-1}) = \int p(x_t, y_t | x_{t-1}, \theta) p(\theta | x_{0:t-1}, y_{1:t-1}) \mathrm{d}\theta \tag{4}$$

where $p(\theta | x_{0:t-1}, y_{1:t-1})$ is the posterior distribution of the parameters. For a general SSM, the integral (4) is intractable, and the posterior may be difficult to compute. However, if there is a conjugacy relationship between the prior distribution $p(\theta)$ and the complete data likelihoods $p(x_{0:t}, y_{1:t} | \theta)$, $t = 1, \dots, T$, the integral can be solved analytically and the posterior will be of the same form as the prior. One such case is when both the complete data likelihood and the parameter prior are in the *exponential family*, see Supplementary A for details. However, if we consider joint state and observation likelihoods, $p(x_t, y_t | x_{t-1}, \theta)$, in the exponential family, we can end up with a log-partition function that depends on the previous state $x_{t-1}$. This can create problems when formulating a conjugate prior for the complete data likelihoods since the prior will be different for each state update, see Supplementary B for details. To avoid this problem for the models we consider, we introduce the *restricted exponential family* where the joint state and observation likelihood is given by

$$p(x_t, y_t | x_{t-1}, \theta) = h_t \exp\left(\theta^\mathsf{T} s_t - A^\mathsf{T}(\theta) r_t\right) \tag{5}$$

where $h_t = h(x_t, x_{t-1}, y_t)$ is the data dependent base measure, $s_t = s(x_t, x_{t-1}, y_t)$ is a sufficient statistic and where the log-partition function can be separated into two factors: $A(\theta)$, which is independent of the data, and $r_t = r(x_{t-1})$, which is independent of the parameters. A conjugate prior for (5) is given by

$$\pi(\theta | \chi_0, \nu_0) = g(\chi_0, \nu_0) \exp\left(\theta^\mathsf{T} \chi_0 - A^\mathsf{T}(\theta)\nu_0\right) \tag{6}$$

where $\chi_0$, $\nu_0$ are hyperparameters. The parameter posterior is given by $p(\theta | x_{0:t-1}, y_{1:t-1}) = \pi(\theta | \chi_{t-1}, \nu_{t-1})$, with the hyperparameters iteratively updated according to

$$\chi_t = \chi_0 + \sum_{k=1}^{t} s_k = \chi_{t-1} + s_t, \qquad \nu_t = \nu_0 + \sum_{k=1}^{t} r_k = \nu_{t-1} + r_t. \tag{7}$$

With the joint likelihood (5) and its conjugate prior (6) in place, we can derive an analytic expression for the marginal of the joint distribution of states and observations, (4), at time $t$

$$p(x_t, y_t | x_{0:t-1}, y_{1:t-1}) = \int p(x_t, y_t | x_{t-1}\theta) \pi(\theta | \chi_{t-1}, \nu_{t-1}) \mathrm{d}\theta = \frac{g(\chi_{t-1}, \nu_{t-1})}{g(\chi_t, \nu_t)} h_t. \tag{8}$$

Hence, to compute the weights (3) for marginalized SMC in the restricted exponential family, we only need to keep track of and update the hyperparameters according to (7).

### 3.2 Marginalized particle Gibbs

In PG we alternate between sampling the parameters and the states like in the hypothetical Gibbs sampler, but the state trajectory is sampled using conditional SMC (cSMC). In cSMC one particle trajectory, the reference trajectory $x'_{0:T}$, will always survive the resampling step. This version of SMC follows the steps in Algorithm 1, with the constraints that $a_t^N = N$ and $x_t^N = x'_t$ (for details, see [2]). When marginalizing out the parameters, the resulting mPG sampler updates the state trajectory using marginalized cSMC (mcSMC), according to what is presented in Algorithm 1 and Section 3.1, with the addition of conditioning on the reference trajectory surviving the resampling step (like in standard PG).

The conditioning used in cSMC yields a Markov kernel that leaves the correct conditional distribution invariant for any choice of $N$ [2]. PG is therefore a valid MCMC procedure. However, it has been shown that $N$ must increase (at least) linearly with $T$ for the kernel to mix properly for large $T$, resulting in an overall computational complexity which grows quadratically with $T$. This holds also for other popular PMCMC methods, such as particle marginal Metropolis-Hastings [2]. To mitigate this issue, [20] proposed a modification of PG in which the ancestor for the reference trajectory in each time step is sampled, according to ancestor weights $\tilde{w}_{t-1|T}^i$, instead of set deterministically, which significantly improves the mixing of the kernel for small $N$, even when $T$ is large. The resulting method, referred to as PGAS, is equivalent to PG apart from the resampling step.

The difference between mPG and mPGAS lies, analogous to the non-marginalized case, only in the resampling step. Deriving the expression for the ancestor weights in the marginalized case is quite involved, below we simply state the necessary expressions and updates, a complete derivation is provided in Supplementary C. Each ancestor trajectory in mPGAS is assigned a weight, based on the general expression in [20], given by

$$\tilde{w}^i_{t-1|T} = \bar{w}^i_{t-1} \frac{\gamma_T([x^i_{0:t-1}, x'_{t:T}])}{\gamma_{t-1}(x^i_{0:t-1})} = \bar{w}^i_{t-1} \frac{p([x^i_{0:t-1}, x'_{t:T}], y_{1:T})}{p(x^i_{0:t-1}, y_{1:t-1})}, \tag{9}$$

where $\bar{w}^i_{t-1}$ is the weight of the ancestor trajectory $x^i_{0:t-1}$ and $[x^i_{0:t-1}, x'_{t:T}]$ is the concatenated trajectory resulting from combining the reference trajectory $x'_{t:T}$ with the possible ancestral path $x^i_{0:t-1}$. For members of the restricted exponential family we use (8) in (9) to get the weights

$$\tilde{w}^i_{t-1|T} \propto \bar{w}^i_{t-1} h^i_t \frac{g(\chi^i_{t-1}, \nu^i_{t-1})}{g(\chi^i_t, \nu^i_t)} \prod_{k=t+1}^{T} h'_k \frac{g(\chi^i_{k-1}, \nu^i_{k-1})}{g(\chi^i_k, \nu^i_k)} \propto \bar{w}^i_{t-1} \frac{g(\chi^i_{t-1}, \nu^i_{t-1})}{g(\chi^i_T, \nu^i_T)} h^i_t, \tag{10}$$

where $\chi^i_{t-1}, \nu^i_{t-1}$ are given, for each particle, by (7) and where

$$\chi^i_T = \chi^i_{t-1} + s_t(x'_t, x^i_{t-1}, y_t) + s'_{t+1:T}, \qquad \nu^i_T = \nu^i_{t-1} + r_t(x^i_{t-1}) + r'_{t+1:T}, \tag{11}$$

with $s'_{t+1:T} = \sum_{k=t+1}^{T} s_k(x'_k, x'_{k-1}, y_k)$ and similarly for $r_t$. Hence, $\chi^i_T$ is a combination of the statistic for the ancestor trajectory, a cross-over term and the statistic for the reference trajectory, which in each timestep is updated according to $s'_{t+1:T} = s'_{t:T} - s_t(x'_t, x'_{t-1}, y_t)$, and analogously for $\nu^i_T$ and $r'_{t+1:T}$. By storing and updating these parameters and sum of statistics in each iteration, computing the ancestor sampling weights only amounts to evaluating (10), implying that we can run mPGAS in linear time despite having a non-Markovian target, which would normally yield quadratic complexity (see [20] for a discussion). We outline mPGAS in Algorithm 2 (for mPG, skip step 3, updates of $\chi_T$, $\nu_T$ and set $a^N_t$ deterministically).

---

**Algorithm 2** Marginalized PGAS for the restricted exponential family (all steps for $i = 1, \ldots, N$)

**Input:** $x'_{0:T}, s'_{1:T}, r'_{1:T}$

1: *Initialize:* Draw $x^{1:N-1}_0 \sim q_0(x_0)$, set $x^N_0 = x'_0$, set $w^i_0 = \frac{\gamma_0(x^i_0)}{q_0(x^i_0)}$ and $\bar{w}^i_0 = \frac{w^i_0}{\sum_{j=1}^{N} w^j_0}$
2: **for** $t = 1 \ldots T$ **do**
3:     *Update statistics:* $s'_{t+1:T} = s'_{t:T} - s_t(x'_t, x'_{t-1}, y_t)$, $r'_{t+1:T} = r'_{t:T} - r_t(x'_{t-1})$
4:     *Update hyperparameters:* $\chi^i_t, \nu^i_t, \chi^i_T, \nu^i_T$ according to (7) and (11)
5:     *Resample:* Draw $a^{1:N-1}_t \sim \mathcal{C}(\{\bar{w}^i_{t-1}\}^N_{i=1})$ and $a^N_t \sim \mathcal{C}(\{\tilde{w}^i_{t-1|T}\}^N_{i=1})$, $\tilde{w}^i_{t-1|T}$ from (10)
6:     *Propagate:* Simulate $x^{1:N-1}_t \sim q_t(x_t|x^{a^{1:N-1}_t}_{0:t-1})$ and set $x^N_t = x'_t$
7:     *Update weights:* Set $w^i_t = \omega_t(x^i_{0:t})$ according to (3) and normalize $\bar{w}^i_t = w^i_t / \sum_{j=1}^{N} w^j_t$
8: **end for**
**Output:** Sample new $x'_{0:T}, s'_{1:T}, r'_{1:T}$ according to $\bar{w}_T$

---

To illustrate the improved performance offered by marginalization we consider the non-linear SSM [15]

$$x_t = \frac{x_{t-1}}{2} + 25 \frac{x_{t-1}}{1 + x^2_{t-1}} + 8\cos(1.2t) + v_t, \qquad y_t = \frac{x^2_t}{20} + w_t, \tag{12}$$

where $v_t$ and $w_t$ are Gaussian noise processes with zero mean and unknown variances $\sigma^2_v$ and $\sigma^2_w$ respectively. The observations are a quadratic function of the state, which makes the posterior multimodal. We will assume conjugate, inverse gamma priors $\sigma^2_v \sim \mathcal{IG}(\alpha_v, \beta_v)$ and $\sigma^2_w \sim \mathcal{IG}(\alpha_w, \beta_w)$ for the unknown variances, with hyperparameters $\alpha_v = \beta_v = \alpha_w = \beta_w = 1$. We generated $T = 150$ observations from (12) with $\sigma^2_v = 10$ and $\sigma^2_w = 1$. PGAS and mPGAS were run for $M = 10000$ iterations, discarding the first 1500 samples as burn-in. We initialized with $\sigma^2_v = \sigma^2_w = 100$ and used a bootstrap proposal for PGAS and a marginalized bootstrap proposal for mPGAS.

Figure 1 shows the autocorrelation for PGAS and mPGAS for different number of particles $N$. Ideally we would like iid samples from the posterior distribution, in terms of the ACF of the samples it

should be zero everywhere except for lag 0. It is clear that, for PGAS, increasing $N$ can reduce the autocorrelation only to a certain limit (given by the hypothetical Gibbs sampler). For mPGAS on the other hand, we obtain a lower autocorrelation using only 50 particles as compared to 5000 for PGAS, and by increasing $N$ we move towards generating iid samples. In Supplementary E we provide corresponding results for PG/mPG. The results in Figure 1 were obtained from our implementation in Matlab, in Supplementary E we also show the corresponding results for our implementation in Birch.

The marginalized versions of PG/PGAS requires some extra computations compared to their non-marginalized counterparts, however, this overhead is quite small. For the model (12), with N=500, using the tic-toc timer in MATLAB we get: PG – 1231.5s, mPG – 1430.7s, PGAS – 1260.7s, mPGAS – 1566.1s. Note that the code has not been optimized.

## 4 Extensions and numerical simulations

In this section we describe three extensions of the marginalized method presented in Section 3 and illustrate their efficiency in numerical examples.

### 4.1 Diffuse priors and blocking

When we do not know much about the parameters of a model, we may use a diffuse prior to reflect our uncertainty. However, a diffuse prior on the parameters can lead to a diffuse prior also for the states. We can then encounter problems during the first few timesteps of the marginalized state trajectory update; in particular, if we use a bootstrap-style proposal in the mcSMC algorithm it may spread out the particles too much. This can result in poor mixing during the initial timesteps, as well as numerical difficulties in the computation of the ancestor sampling weights, due to very large values sampled for the states. As an illustration, consider again the model (12), but now with hyperparameters $\alpha_v, \beta_v = 0.001$ for the process noise $\sigma_v^2$. The marginalized proposal for the first timestep, $q_1(x_1|x_0) = p(x_1 \mid x_0)$, will then be a Student $t$-distribution with undefined mean and variance. Figure 2 (left) shows the log-pdf of both this proposal and the target distribution, $\bar{\gamma}_1(x_{0:1}) = p(x_{0:1} \mid y_1)$, at time $t = 1$. It is clear that for mcSMC (blue) the prior $q_1$ is much more diffuse than the posterior $\bar{\gamma}_1$, whereas for cSMC (orange) there is less of a difference.

When working with diffuse priors we suggest to divide the state trajectory into two overlapping blocks (similarly to the blocking method proposed by [34]) and do Gibbs updates of each block in turn. Figure 3 illustrates the two overlapping blocks $x_{0:B+L}$ (upper) and $x_{B+1:T}$ (lower). To update the first block, where problems due to marginalization are more probable, we use a (non-marginalized) cSMC sampler targeting the posterior distribution of $x_{0:B+L}$ conditioned on the reference trajectory $x'_{0:B+L}$, the observations $y_{1:T}$, the non-overalapping part of the second block $x'_{B+L+1:T}$ and the parameters $\theta$. Note that, because of the Markov property when conditioning on $\theta$, the dependence on $x_{0:B+L}$ reduces to only the boundary state $x'_{B+L+1}$ and the dependence on the observations reduces to $y_{1:B+L}$. To update the second block, we use mcSMC targeting the posterior distribution of $x_{B+1:T}$ conditioned on the (updated) reference trajectory $[x_{B+1:B+L}, x'_{B+L+1:T}]$, the observations $y_{1:T}$ and the (updated) first block $x_{0:B}$. Finally, the parameters $\theta$ are sampled from their full conditional given the new reference

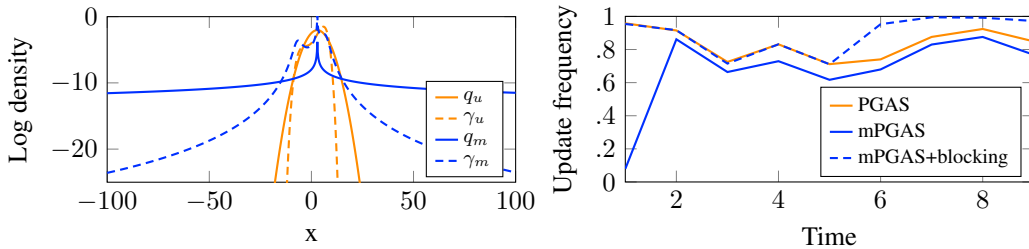

Figure 2: *Left:* log-density for the proposal and the posterior at $t = 1$ for mcSMC ($q_m$, $\gamma_m$) and cSMC ($q_u$, $\gamma_u$), showing how marginalization can potentially produce a poor proposal distribution in the first timestep. *Right:* update frequency for the state trajectory for the first few timesteps.

trajectory $x_{0:T}$. Algorithm 3 outlines one iteration of mPG/mPGAS with this choice of blocking and samplers. In Supplementary D we provide a proof of validity for this blocked Gibbs sampler.

The purpose of the first block is only to update the first few timesteps, in order to get a sufficient concentration of the proposals when conditioning on $x_{0:B}$ for mcSMC. Therefore, it is typically sufficient to use a small value of $B$; in the example outlined above $B > 2$ is sufficient to get finite variance in the Student $t$-distribution. The overlap parameter $L$ on the other hand is used to push the boundary state $x_{B+L+1}$ into the interior of the second block, which due to the forgetting of the dynamical system reduces the effect of conditioning on this state in the first Gibbs step [34]. Hence, the larger $L$ the better, but at the price of increased computational cost. Since most SSMs have exponential forgetting, using a small value of $L$ is likely to be sufficient in most cases.

In Figure 2 (right), we illustrate the benefit of using blocking to avoid poor mixing during the first timestep when marginalizing with a diffuse prior for the model (12). We used $B = 5$ and $L = 20$, all other settings were the same as before. We consider the update frequency of the state variables, defined as the average number of iterations in which the state changes its value, as a measure of the mixing. It is clear that for the mPGAS we get a very low update frequency at $t = 1$, whereas when we use mPGAS with blocking we obtain the same update frequency as for PGAS.

| **Algorithm 3** Blocking for mPG/mPGAS |
| --- |
| 1: $x_{0:B+L} \sim \text{cSMC}(x_{0:B+L}\|x'_{0:B+L}; y_{1:B+L}, x'_{B+L+1}, \theta)$ |
| 2: $x_{B+1:T} \sim \text{mcSMC}(x_{B+1:T}\|x_{B+1:B+L}, x'_{B+L+1:T}; x_{0:B}, y_{1:T})$ |
| 3: $\theta \sim p(\theta\|x_{0:T}, y_{1:T}) = \pi(\theta\|\chi_T, \nu_T)$ |

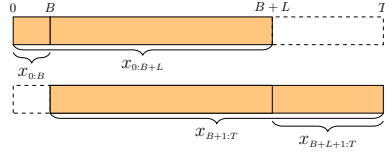

Figure 3: Division into 2 blocks.

## 4.2 Marginalized particle Gibbs in a PPL

We have implemented PG, PGAS, mPG and mPGAS in Birch [24], which employs *delayed sampling* [26] to recognize and utilize conjugacy relationships, and so automatically marginalizes out the parameters of a model, where possible. This saves the user the trouble of deriving the relevant conjugacy relationships for their particular model, or providing a bespoke implementation of them. We first demonstrate this on a vector-borne disease model of a dengue outbreak.

Dengue is a mosquito-borne disease which affects an estimated 50-100 million people worldwide each year, causing 10000 deaths [35]. We use a data set from an outbreak on the island of Yap in Micronesia in 2011. It contains 197 observations, mostly daily, of the number of newly reported cases. The model used is that described in [26], in turn based on that of the original study [12]. It consists of two coupled susceptible-exposed-infectious-recovered (SEIR) compartmental models, describing the transmission between human and mosquito populations, respectively. Transition counts between compartments are assumed to be binomially distributed, with beta priors used for all parameters. Observations are also assumed binomial with an unknown parameter for the reporting rate, which is assigned a beta prior. The beta priors establish conjugate relations with the complete data likelihood, so that the problem is well-suited for inference using mPG/mPGAS.

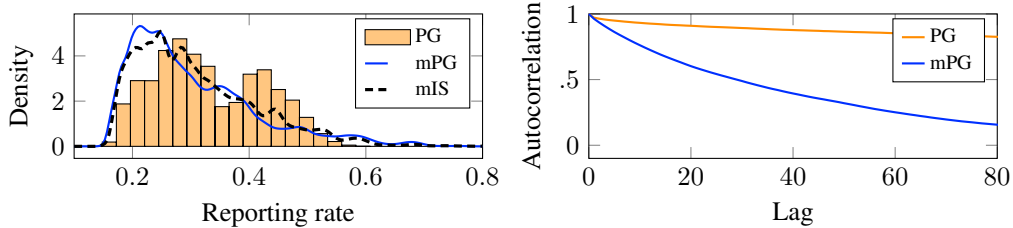

Figure 4: Results of the simulation of the vector-borne disease model. *Left:* estimated density of the reporting rate parameter, mean of four chains. Marginalized importance sampling (mIS) is included for comparison. *Right:* estimated autocorrelation function of the reporting rate parameter, mean of four chains.

The model was previously implemented in Birch for [26]. We have added generic implementations of PG, PGAS, mPG and mPGAS to Birch that can be applied to this, and other, models. Figure 4 shows the results of a simulation of four different chains; for each of these 10000 samples were drawn using PG and mPG. The samplers used $N = 1024$ particles each. For comparison we also include the results from using marginalized importance sampling (mIS). The autocorrelation of the samples is noticeably improved by marginalizing out the parameters. Corresponding results for PGAS and mPGAS can be found in Supplementary E.

## 4.3 Models lacking full conjugacy

It may seem that the method we propose is limited to models where the transition and observation probabilities have the conjugacy structure in (5). However, we can use the results in Section 3 to treat models where only some of the parameters exhibit conjugacy with the complete data likelihood. To this end, we denote by $\theta_m$ the parameters that have a prior distribution that is conjugate with the complete-data likelihood, and by $\theta_u$ the remaining parameters. Then, we can marginalize out $\theta_m$ from the complete-data likelihood as shown in Section 3. The remaining parameters can be sampled using any conventional MCMC method, for instance Metropolis–Hastings. This is possible since PMCMC samplers are nothing but (special purpose) MCMC kernels, hence they can be combined with normal MCMC in a systematic way. One possibility is to use, say, Metropolis–Hastings within mPG/mPGAS. Another possibility, which we describe below, is to use a marginalized version of the particle marginal Metropolis–Hastings algorithm [2], which we refer to as mPMMH.

Let $\hat{p}(y_{1:T}|\theta_u) = \prod_{t=1}^{T} \frac{1}{N} \sum_{i=1}^{N} w_t^i$ be the unbiased estimate of the marginal likelihood given by Algorithm 1, for a fixed value of the parameters $\theta_u$, and let $q(\theta_u|\theta'_u)$ be a proposal distribution; then, we can generate samples from the posterior distribution of $\theta_u$ using Algorithm 4.

---

**Algorithm 4** Marginalized particle marginal Metropolis–Hastings

1: Propose $\theta_u^* \sim q(\,\cdot\,|\theta'_u)$
2: Run Algorithm 1 and compute $\hat{p}(y_{1:T}|\theta_u^*)$
3: Return $\theta_u = \theta_u^*$ with probability $1 \wedge \frac{\hat{p}(y_{1:T}|\theta_u^*)p(\theta_u^*)q(\theta'_u|\theta_u^*)}{\hat{p}(y_{1:T}|\theta'_u)p(\theta'_u)q(\theta_u^*|\theta'_u)}$, else $\theta_u = \theta'_u$,

---

To illustrate this method with partial marginalization, we consider the following model describing the evolution of the size of animal populations (see, for instance, [17]):

$$\log n_{t+1} = \log n_t + \begin{bmatrix} 1 & (n_t)^c \end{bmatrix} b + \sigma_v v_t, \qquad y_t = n_t + \sigma_w w_t,$$

where $n_t$ is the population size at time $t$, and $b$, $c$, $\sigma_v$, and $\sigma_w$ are the unknown parameters. Note that, except for $c$ ($= \theta_u$), the parameters can be marginalized out by using normal-inverse gamma and inverse gamma conjugate priors $b, \sigma_v^2 \sim \mathcal{NIG}(\mu, \Lambda, \alpha_v, \beta_v)$ and $\sigma_w^2 \sim \mathcal{IG}(\alpha_w, \beta_w)$. For the remaining parameter, we use a $\mathcal{N}(0, \sigma_c^2)$ prior and a random-walk proposal $c^* \sim \mathcal{N}(c', \tau)$.

We have implemented mPMMH in Birch and evaluate it on a dataset of observations of the number of song sparrows on Mandarte Island, British Columbia, Canada [33]. The dataset contains the number of birds, counted yearly, between 1978 and 1998. In Figure 5 (left), we report the histogram of the distribution of the density regulation parameter $c$ estimated using 10000 samples drawn using Algorithm 4 after a burn-in of 5000 samples, using $N = 512$ particles. The distribution of $c$, as found by our method, is consistent with values reported in the literature (see, for instance, [27] and [33]). In Figure 5 (right), we show the actual counts in the dataset compared with the average $\hat{n}_{1:T}$ and three standard deviations, as sampled by Algorithm 4.

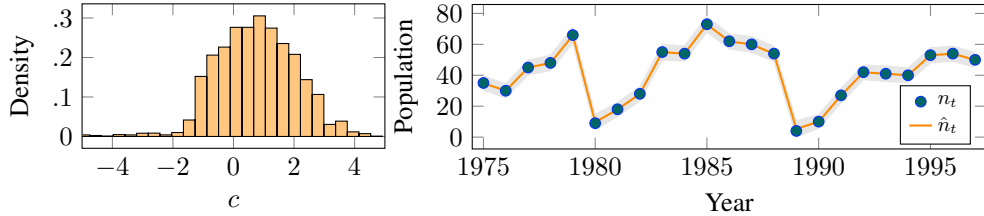

Figure 5: Results of the simulation with parameter values $\mu = [1, 1]$, $\Lambda = I$, $\alpha_v = \beta_v = \alpha_w = \beta_w = 2.5$, $\sigma_c^2 = 4$, $\tau = 0.05$. *Left:* estimated distribution of the density regulation parameter $c$. *Right:* observed (marks) and mean filtered population sizes (solid) with $3\sigma$ credible interval.

## 5 Discussion

PG and PGAS can be highly efficient samplers for general SSMs, but are limited by the performance of the hypothetical (but intractable) Gibbs sampler that they approximate. We have proposed to improve on PG/PGAS by marginalizing out the parameters from the state update, to reduce the auto-correlation beyond the limit posed by the hypothetical Gibbs sampler.

Marginalization often improves performance, but this will not always be the case. One example is when there is a diffuse prior on the parameters, in which case marginalization can result in an inefficient SMC sampler. One way to mitigate this is blocking; we propose using two blocks, the first updated using cSMC and the second using mcSMC. One can think of other ways to update the first block, such as a Metropolis–Hastings update with an appropriate proposal, see [11, 25] for related techniques. It is also possible to use a mcSMC update for the first block, as conditioning on the future states will help to avoid the problems related to diffuse priors. The details are quite involved, however, so we prefer the simpler method described in Section 4.1.

Marginalization is possible when there is a conjugacy relationship between the parameters and the complete data likelihood. This may seem a restrictive model class, but in practice there are benefits even if only some of the parameters can be marginalized out, by combining marginalized PMCMC kernels with conventional MCMC kernels. Many models have at least some parameters that enter in a nice way, such as regression coefficients and error variances, where marginalization can provide a performance gain.

Performing the marginalization by hand for every new model can be time consuming. Consequently, an important aspect of the method is the possibility of implementing it in a probabilistic programming language. Recent advances in probabilistic programming enable automatic marginalization, making the process easier. We have implemented mPG, mPGAS and mPMMH in Birch, and demonstrated that implementation on two examples. Some further work is required to extend the implementation in Birch to blocking.

**Code**

Code for all numerical simulations is available at `https://github.com/uu-sml/neurips2019-parameter-elimination`.

**Acknowledgments**

This research is financially supported, in part, by the Swedish Research Council via the projects *Learning of Large-Scale Probabilistic Dynamical Models* (contract number: 2016-04278) and *New Directions in Learning Dynamical Systems (NewLEADS)* (contract number:2016-06079), by the Swedish Foundation for Strategic Research (SSF) via the projects *Probabilistic Modeling and Inference for Machine Learning* (contract number: ICA16-0015) and *ASSEMBLE* (contract number: RIT15-0012), and by the Wallenberg AI, Autonomous Systems and Software Program (WASP) funded by the Knut and Alice Wallenberg Foundation.

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
