[Supplementary Material]

# Supplementary material

## A  Exponential family

A generic exponential family distribution can be written

$$p(z|\eta) = h(z)\exp\left(\eta^\mathsf{T} s(z) - a(\eta)\right) \tag{13}$$

where $h$ is the data dependent base measure, $a$ is the log-partition function, and $s$ is a sufficient statistic storing all the information about the natural parameters $\eta$ contained in the data $z$. The conjugate prior for an exponential family distribution is also in the exponential family and is given by

$$\pi(\eta|\chi, \nu) = g(\chi, \nu)\exp\left(\eta^\mathsf{T}\chi - a(\eta)\nu\right) \tag{14}$$

where $g$ is a normalizing factor and $\chi$, $\nu$ are hyperparameters. The parameter posterior for a prior (14) and a likelihood (13) is given by

$$p(\eta|z, \chi, \nu) \propto p(z|\eta)p(\eta|\chi, \nu) = h(z)g(\chi, \nu)\exp\left(\eta^\mathsf{T}\left(\chi + s(z)\right) - a(\eta)(\nu + 1)\right) \tag{15}$$

where Bayes' rule was used in the first proportionality. If we compare the exponential factor in the posterior (15) with the prior (14) we note that the posterior indeed is of the same form as the prior, but with updated hyperparameters $\chi_{\text{new}} = \chi + s(z)$ and $\nu_{\text{new}} = \nu + 1$. Hence, we have conjugacy for distributions in the exponential family, and the parameter posterior is given by $p(\eta|z, \chi, \nu) = \pi(\eta|\chi + s(z), \nu + 1)$. We can obtain the likelihood of the data $z$ by marginalizing out the natural parameters $\eta$ from the joint distribution $p(z, \eta|\chi, \nu) = p(z|\eta)p(\eta|\chi, \nu)$ which gives

$$p(z|\chi, \nu) = \int p(z, \eta|\chi, \nu)\mathrm{d}\eta = h(z)g(\chi, \nu)\int \underbrace{\exp\left(\eta^\mathsf{T}\left(s(z) + \chi\right) - a(\eta)\left(1 + \nu\right)\right)}_{\text{Unnormalized } \pi(\eta|\chi_{\text{new}}, \nu_{\text{new}})}\mathrm{d}\eta$$

$$= h(z)\frac{g(\chi, \nu)}{g(\chi_{\text{new}}, \nu_{\text{new}})}$$

where (14) and (13) were inserted in the second equality. Hence, for members of the exponential family there is a closed form expression for the distribution of the data when the parameters have been marginalized out.

## B  Restricted exponential family

When working with SSMs we can run into problems when using the standard formulation for likelihoods and priors for the exponential family presented in Supplementary A. The reason for this is that we typically have a dependence on the previous state $x_{t-1}$ in the transition density that can result in a log-partition function which depends on this previous state. Put in the same general notation as in the previous section we get the likelihood

$$p(z|\zeta, \eta) = h(z, \zeta)\exp\left(\eta^\mathsf{T} s(z, \zeta) - a(\eta, \zeta)\right)$$

where the variable $\zeta$ is a known extra parameter (e.g the previous state $x_{t-1}$) and we note that the log-partition function $a$ depends on this parameter. If we wish to formulate an exponential family prior for this likelihood we get

$$\pi(\eta|\chi, \nu, \zeta) = g(\chi, \nu)\exp\left(\eta^\mathsf{T}\chi - a(\eta, \zeta)\nu\right),$$

that is, we get a different prior depending on the value on $\zeta$ and, consequently, we cannot easily formulate a general prior distribution which is conjugate to the complete data likelihood. Instead we propose to use a restricted exponential family where the log-partition function is assumed to be separable into one parameter-dependent part and one $\zeta$-dependent part. The likelihood is given by

$$p(z|\zeta, \eta) = h(z, \zeta)\exp\left(\eta^\mathsf{T} s(z, \zeta) - A^\mathsf{T}(\eta)r(\zeta)\right)$$

where $A(\eta)$ is the restricted log-partition function and $r(\zeta)$ is some function which only depends on $\zeta$. A conjugate prior for this likelihood is

$$\pi(\eta|\chi, \nu, \zeta) = g(\chi, \nu)\exp\left(\eta^\mathsf{T}\chi - A^\mathsf{T}(\eta)\nu\right).$$

The parameter posterior is

$$\pi(\eta|z,\zeta,\chi,\nu) \propto p(z|\zeta,\eta)p(\eta|\chi,\nu,\zeta) = h(z,\zeta)g(\chi,\nu)\exp\left(\eta^\mathsf{T}\left(\chi+s(z,\zeta)\right)-A^\mathsf{T}(\eta)(\nu+r(\zeta))\right).$$

We note that the posterior is of the same form as the prior but with updated hyperparameters $\chi_{\text{new}} = \chi + s(z,\zeta)$ and $\nu_{\text{new}} = \nu + r(\zeta)$. Comparing with the standard exponential family we note that the only difference is that the statistic now depends also on $\zeta$ and that we get a statistic, $r(\zeta)$, to update also for $\nu$.

## C Derivation of ancestor weights for the restricted exponential family

For SSMs with joint state and observation likelihood in the restricted exponential family the likelihood is given by (5) and the parameter prior is $\pi(\theta|\chi_0,\nu_0) = g(\chi_0,\nu_0)\exp\left(\theta^\mathsf{T}\chi_0 - A^\mathsf{T}(\theta)\nu_0\right)$. The ancestor weights are given by equation (9), which can be expanded to

$$
\begin{aligned}
\tilde{w}^i_{t-1|T} &\propto \bar{w}^i_{t-1}\frac{p(x'_{t:T},y_{t:T}|x^i_{0:t-1},y_{1:t-1})p(x^i_{0:t-1},y_{1:t-1})}{p(x^i_{0:t-1},y_{1:t-1})} = \bar{w}^i_{t-1}p(x'_{t:T},y_{t:T}|x^i_{0:t-1},y_{1:t-1})\\
&= \bar{w}^i_{t-1}p(x'_t,y_t|x^i_{0:t-1},y_{1:t-1})\prod_{k=t+1}^{T}p(x'_k,y_k|x^i_{0:t-1},y_{1:t-1},x'_{t:k-1},y_{t:k-1})\\
&= \bar{w}^i_{t-1}\int p(x'_t,y_t|x^i_{t-1},\theta)p(\theta|x^i_{0:t-1},y_{1:t-1})\mathrm{d}\theta\\
&\qquad \prod_{k=t+1}^{T}\int p(x'_k,y_k|x'_{k-1},\theta)p(\theta|x^i_{0:t-1},x'_{t:k-1},y_{1:k-1})\mathrm{d}\theta.
\end{aligned}
\tag{16}
$$

Now, to continue we need to compute the two integrals in the last equality for members in the restricted exponential family. First, note that

$$p(x^i_{0:t-1},y_{1:t-1}|\theta) = \Big(\prod_{j=1}^{t-1}h^i_j\Big)\exp\Big(\theta^\mathsf{T}\sum_{j=1}^{t-1}s^i_j - A^\mathsf{T}(\theta)\sum_{j=1}^{t-1}r^i_j\Big)$$

and therefore

$$
\begin{aligned}
p(\theta|x^i_{0:t-1},y_{1:t-1}) &\propto p(x^i_{0:t-1},y_{1:t-1}|\theta)p(\theta|\chi,\nu)\\
&= g(\chi_0,\nu_0)\Big(\prod_{j=1}^{t-1}h^i_j\Big)\exp\Big(\theta^\mathsf{T}\big(\chi_0+\sum_{j=1}^{t-1}s^i_j\big) - A^\mathsf{T}(\theta)\big(\nu_0+\sum_{j=1}^{t-1}r^i_j\big)\Big)\\
&\propto \pi(\theta|\chi^i_{t-1},\nu^i_{t-1}),
\end{aligned}
$$

where $\chi^i_{t-1} = \chi_0 + \sum_{j=1}^{t-1}s^i_j = \chi_0 + s^i_{1:t-1}$ and $\nu^i_{t-1} = \nu_0 + \sum_{j=1}^{t-1}r^i_j = \nu_0 + r^i_{1:t-1}$ and the last proportionality comes from comparison with the exponential factor in the prior. Similarly, by splitting in three terms (one for the ancestral part, one for the cross-over and one for the reference trajectory part) we have

$$
\begin{aligned}
p(x^i_{0:t-1},x'_{t:k-1},y_{1:k-1}|\theta) &= \Big(\prod_{j=1}^{t-1}p(x^i_j,y_j|x^i_{j-1},\theta)\Big)p(x'_t,y_t|x^i_{t-1},\theta)\Big(\prod_{j=t+1}^{k-1}p(x'_j,y_j|x'_{j-1},\theta)\Big)\\
&= \underbrace{\Big(\prod_{j=1}^{t-1}h^i_j\Big)h'_t\Big(\prod_{j=t+1}^{k-1}h'_j\Big)}_{H}\exp\Big(\theta^\mathsf{T}\big(\underbrace{s^i_{1:t-1}+s_t(x'_t,x^i_{t-1},y_t)+s'_{t+1:k-1}}_{S}\big)\\
&\qquad - A^\mathsf{T}(\theta)\big(\underbrace{r^i_{1:t-1}+r_t(x^i_{t-1})+r'_{t+1:k-1}}_{R}\big)\Big)
\end{aligned}
$$

and therefore, using the abbreviations $H$, $S$, and $R$ above, the parameter posterior is

$$
\begin{aligned}
p(\theta|x^i_{0:t-1},x'_{t:k-1},y_{1:k-1}) &\propto g(\chi_0,\nu_0)H\exp\Big(\theta^\mathsf{T}\big(\chi_0+S\big) - A^\mathsf{T}(\theta)\big(\nu_0+R\big)\Big)\\
&\propto \pi(\theta|\chi^i_{k-1},\nu^i_{k-1})
\end{aligned}
$$

where

$$\chi_{k-1}^i = \chi_0 + s_{1:t-1}^i + s_t(x_t', x_{t-1}^i, y_t) + s_{t+1:k-1}' = \chi_{t-1}^i + s_t(x_t', x_{t-1}^i, y_t) + s_{t+1:k-1}'$$
$$\nu_{k-1}^i = \nu_0 + r_{1:t-1}^i + r_t(x_{t-1}^i) + r_{t+1:k-1}' = \nu_{t-1}^i + r_t(x_{t-1}^i) + r_{t+1:k-1}'.$$

We can now compute the integrals in (16):

$$\int p(x_t', y_t | x_{t-1}^i, \theta) p(\theta | x_{0:t-1}^i, y_{1:t-1}) \mathrm{d}\theta =$$

$$\int h_t \exp\left(\theta^\mathsf{T} s_t' - A^\mathsf{T}(\theta) r_t'\right) g(\chi_{t-1}^i, \nu_{t-1}^i) \exp\left(\theta^\mathsf{T} \chi_{t-1}^i - A^\mathsf{T}(\theta)\nu_{t-1}^i\right)\mathrm{d}\theta$$

$$= h_t g(\chi_{t-1}^i, \nu_{t-1}^i) \int \underbrace{\exp\left(\theta^\mathsf{T}(s_t' + \chi_{t-1}^i) - A^\mathsf{T}(\theta)(r_t' + \nu_{t-1}^i)\right)}_{\text{Unnormalized } \pi(\theta | \chi_t^i, \nu_t^i)} \mathrm{d}\theta \qquad (17)$$

$$= h_t \frac{g(\chi_{t-1}^i, \nu_{t-1}^i)}{g(\chi_t^i, \nu_t^i)}.$$

Note that $h_t = h_t(x_t', x_{t-1}^i, y_t)$ depends on the ancestral path. In a similar fashion, for the second integral we obtain

$$\int p(x_k', y_k | x_{k-1}', \theta) p(\theta | x_{0:t-1}^i, x_{t:k-1}', y_{1:k-1}) \mathrm{d}\theta =$$

$$h_k' g(\chi_{k-1}^i, \nu_{k-1}^i) \int \exp\left(\theta^\mathsf{T}(s_k' + \chi_{k-1}^i) - A^\mathsf{T}(\theta)(r_k' + \nu_{k-1}^i)\right)\mathrm{d}\theta \qquad (18)$$

$$= h_k' \frac{g(\chi_{k-1}^i, \nu_{k-1}^i)}{g(\chi_k^i, \nu_k^i)},$$

where $h_k' = h_k'(x_k', x_{k-1}', y_t)$ only depends on the reference trajectory and observations. Now, by substituting (17) and (18) into (16), we obtain

$$\tilde{w}_{t-1|T}^i \propto \bar{w}_{t-1}^i h_t \frac{g(\chi_{t-1}^i, \nu_{t-1}^i)}{g(\chi_t^i, \nu_t^i)} \prod_{k=t+1}^T h_k' \frac{g(\chi_{k-1}^i, \nu_{k-1}^i)}{g(\chi_k^i, \nu_k^i)}$$

$$\propto \bar{w}_{t-1}^i h_t \frac{g(\chi_{t-1}^i, \nu_{t-1}^i)}{g(\chi_T^i, \nu_T^i)},$$

where the (surprisingly) simple final expression is due to all terms except $g(\chi_{t-1}^i, \nu_{t-1}^i)$ and $g(\chi_T^i, \nu_T^i)$ canceling in the product. Note also that $h_k'$ can be removed (in proportionality) since they are the same for all ancestor particles.

## D    Theoretical justification of the blocking scheme

To show that the blocking scheme in Algorithm 3 is correct we start by verifying that the underlying (hypothetical) partially collapsed Gibbs (PCG) sampler is correct. Following the notation in [41] we set $X_1 = x_{0:B}$, $X_2 = x_{B+1:B+L}$ (the overlap), $X_3 = x_{B+L+1:T}$, $Y = y_{1:T}$, and the superscript * indicates intermediate quantities. The joint distribution we wish to sample from is $p(X_1, X_2, X_3, \theta | Y)$. The underlying hypothetical PCG of the blocking scheme is

$$X_1, X_2^* \sim p(X_1, X_2 | X_3, Y, \theta)$$
$$X_2, X_3 \sim p(X_2, X_3 | X_1, Y) \qquad (19)$$
$$\theta \sim p(\theta | X_1, X_2, X_3, Y).$$

We note that step 1 and 3 are correct hypothetical Gibbs steps, the only issue is the marginalized step 2. However, in step 2 it is possible to also sample $\theta$ from its full conditional, that is to sample $X_2, X_3, \theta^* \sim p(X_2, X_3 | X_1, Y) p(\theta | X_1, X_2, X_3, Y) = p(X_2, X_3, \theta | X_1, Y)$. This sampling of $\theta^*$ is redundant since the sampled value is never conditioned on in the following step and can be removed according to the reasoning in [41], concluding that the underlying hypothetical PCG of the blocking scheme is indeed a correct hypothetical PCG. If one or more of the conditionals in (19) are not

possible to sample from directly some MCMC scheme (such as PG/mPG) can be used. Describing the PMCMC steps in terms of kernels, $K$, Algorithm 3 can be formulated

$$X_1, X_2^* \sim K\left(X_1, X_2 | X_1', X_2'; X_3, Y, \theta\right)$$
$$X_2, X_3 \sim K\left(X_2, X_3 | X_2^*, X_3'; X_1, Y\right)$$
$$\theta \sim p\left(\theta | X_1, X_2, X_3, Y\right)$$

where $'$ indicates the (reference) trajectory from the previous iteration. Again, step 1 and 3 are correct (as long as $K$ in step 1 is correct, e.g. a PG kernel), the main concern is step 2. However, as was the case for the hypothetical sampler, it is possible to add the sampling of $\theta$ in step 2. Sampling $\theta$ is, again, redundant and step 2 is also correct provided that the stationary distribution of $K$ is the marginalized target.

## E    Additional results

In this section we provide some additional results from numerical simulations.

### E.1    Toy-model in Section 3.2

Figure 6 shows the results for our implementation of (12) in Matlab (left) and Birch (right). We generated $T = 250$ observations from (12) with $\sigma_v^2 = 5.3$ and $\sigma_w^2 = 9$. We used hyperparameters $\alpha_v = \alpha_w = 2$ and $\beta_v = \beta_w = 10$, and all four methods were run for $M = 10000$ iterations. We initialized with $\sigma_v^2 = 4.3$ and $\sigma_w^2 = 9.4$ and used a bootstrap proposal for PG/PGAS and a marginalized bootstrap proposal for mPG/mPGAS. We note that for both implementations there is a clear improvement from marginalizing out the parameters (for both PG and PGAS), and that it is beneficial to use PGAS/mPGAS rather than PG/mPG for this model.

Figure 6: Results of the simulation of the model (12). *Left:* autocorrelation function for all methods, obtained from simulations in Matlab. *Right:* autocorrelation function for all methods, obtained from simulations in Birch.

### E.2    Vector-borne disease model in Section 4.2

Figure 7 shows the results of a simulation of the vector-borne disease model in Birch of four different chains; for each of these 10000 samples were drawn using PGAS and mPGAS. The samplers used $N = 1024$ particles each. The autocorrelation of the samples is improved by marginalizing out the parameters. However, we note that there is no obvious improvement from using PGAS rather than PG for this model. One explanation for this could be that it is difficult to match the reference trajectory with an ancestor which is not the reference trajectory for this compartmental model. The reference trajectory would then be sampled in the ancestor sampling step, which in turn means that PGAS would reduce to PG.

Figure 7: Results of the simulation of the vector-borne disease model. *Left:* estimated density of the reporting rate parameter for PGAS and mPGAS, mean of four chains. For comparison, a run of marginalized importance sampling (mIS) is also shown. *Right:* estimated autocorrelation function of the reporting rate parameter for all methods, four chains for each method.