[Reviews · NeurIPS 2019]

Reviewer 1



The marginalisation of variables within some steps of an MCMC algorithm is delicate. The main proposal here appears well justified, but it would have been nice to see the argument made a little more explicitly. The type of marginalisation described here seems to be more or less what would be described as a (partially) collapsed Gibbs sampler in the sense of [David A Van Dyk and Taeyoung Park. “Partially collapsed Gibbs samplers: Theory and methods”. In: Journal of the American Statistical Association 103.482 (2008), pp. 790–796] and I would have liked to see that connection made -- particularly as this is one of the arguments that can be used to justified the basic particle Gibbs sampler. It was less clear to me exactly how the "blocking" strategy detailed in Section 4.1 would be justified from a formal perspective, and I do think that this needs clarifying. I.e. the collection of variables to be sampled is divided into three parts -- x', x~ and theta and the decomposition of the kernel seems to involve sampling: x~ from a kernel invariant to its distribution conditional on both x' and theta (starting from the previous x~) x' from a kernel invariant with respect to its distribution conditional only upon x~ (starting from the previous x') \theta from its full conditional distribution and it's not completely transparent how one knows that this is invariant with respect to the correct joint distribution. In the numerical study I would have liked to see some sort of indication of how algorithmic parameters (especially N) were specified, and some illustration of the dependence on N of the results obtained. I would have also liked to see some kind of explicit statement about computational cost, even if only in terms of wallclock time, as there seem to be additional computations to do in the marginalized case: is the cost of running mPG the same as that as running PG for a given number of particles? I was disappointed not to see PGAS featuring in the comparison because this is also known to dramatically reduce the autocorrelation in many settings; in particular it would have been useful to know (a) how does the improvement arising from using mPG rather than PG relate to that obtained from using PGAS rather than PG and especially if one is already using PGAS then does one observe significant improvement by using mPGAS. Details: line 47: are particle Gibbs samplers really pseudomarginal algorithms? They don't seem obviously to fit the framework of [1], but certainly work as simple Gibbs samplers on a `demarginalisation' of the original target and indeed, wouldn't be expected to outperform the ideal algorithm. line 45-56: I find this discussion a bit misleading. It's possible to outperform what you call the "ideal" Gibbs sampler by implementing a sampler which draws iid samples from the posterior (sampling the state sequence from its marginal posterior and then the parameters from the full conditional distribution amounts to using one particular decomposition by which one could sample from the posterior distribution, at least in principle). It's this idealised algorithm which you seek to approximate with the methods described here, and presumably you would not expect to outperform that algorithm... Section 3.1. It might help the reader if you explain what is meant by a "general SSM". In most of the literature the term is used to refer to Markovian models but that seems not to be what is intended here. line 126: "se" line 130: a conjugate prior is stated for computational convenience; it would be nice if any comment on the interpretation of this prior or its desirability for mode Figure 4: it's not clear to me why a kernel density estimate is shown for one algorithm but a histogram is given for the other. This just seems to prevent direct comparison. Note on 7, below: I understand that code will be provided should the manuscript be accepted. -- I thank the authors for their response and, particularly, for the clarification of the justification of the blocking strategy. Together with the other referee reports this leaves me more strongly in favor of accepting the manuscript.

Reviewer 2



This is a nice paper, potentially giving very important improvements for certain models. Can the ideas also be implemented in a backwards-sampling implementation of the SMC algorithm, rather than ancestor sampling? Is there any understanding of how the degeneracy of the sufficient statistics (as reported in various previous attempts to marginalize out static parameters) affects the algorithm? Is it that this is essentially mitigated by the ancestor sampling? Having read the authors' response and the other reviews, I remain of the opinion that this is a nice contribution. There are always unanswered questions from good papers, especially when they are necessarily short, so it is not surprising that there are some here as well.

Reviewer 3



The paper introduces several novel variants of existing approaches that seem to be of high relevance to me. The description of the novel approaches is generally very clear. The paper is of high quality and I highly appreciate that the authors looked into short-comings of their approach. I would be nice to extend the evaluation with more large models and on more datasets, which should be rather easy as the implementation is done in a PPL. Further, I would appreciate if the authors would also compare the running times. However, the paper sometimes lacks a good explanation of the math used. In particular, (1) Before equation 3 the authors state that the unnormalized target density can be factorized. However, the index k never appears in the equation. I suppose the is a typo. (2) Equation 5 lacks sufficient details to understand it well. (At least in my case). What is h, theta, s, A and r? It would be good if the authors would improve this paragraph. (2) In Equation 8 the weight tilde(w) is not introduced. Further, the paper states that several times that a certain property is illustrated in Figure 1. I was missing a proper explanation of what I should be able to see in Figure 1. I would suggest improving those parts. And last but not least, the authors write that PGAS is not affected by path degeneracy (line 81). However, according to F. Lindsten, P. Bunch, S.S. Singh and T.B. Schön (2015) Particle ancestor sampling for near-degenerate or intractable state transition models this doesn't seem to be 100% correct. ---- I want to thank the authors for their response letter.

[Author Response · NeurIPS 2019]

We want to thank all the reviewers for providing helpful questions and comments. Here we comment on and clarify

some of your questions/concerns, but all comments will of course be addressed when we revise the paper.

**Reviewer #1**

The connection with PCG samplers and the formal justification of the blocking strategy: It is possible to motivate

the marginalized versions of PG/PGAS using the PCG framework of [2], but in the basic setting when $\theta$ is fully

marginalized, this is not necessary. Simply using cSMC on the marginalized target for the state trajectory will yield

samples from the correct stationary distribution. We do, however, use the PCG framework for motivating the blocking

strategy. To see that this is correct, set $X_1 = x_{0:B}$, $X_2 = x_{B+1:B+L}$, $X_3 = x_{B+L+1:T}$, $Y = y_{1:T}$. The block sampler

essentially implements the following Gibbs sweep to sample from $p\left(X_1, X_2, X_3, \theta | Y\right)$

$$X_1, X_2^* \sim p\left(X_1, X_2 | X_3, Y, \theta\right), \quad X_2, X_3 \sim p\left(X_2, X_3 | X_1, Y\right), \quad \theta \sim p\left(\theta | X_1, X_2, X_3, Y\right).$$

Step 1 and 3 are standard Gibbs steps. In step 2 , $\theta$ is collapsed, which can be thought of as adding a draw of $\theta$ from its

full conditional to yield a draw from $p\left(X_2, X_3, \theta | X_1, Y\right)$. Since $\theta$ is not conditioned on in step 3, removing $\theta$ from

step 2 is valid and the scheme is correct. Similar arguments can be applied for the actual sampler, which makes use of

cSMC kernels in step 1 and 2. We agree that the validity of the block sampler was not very clearly shown in the paper,

and we have therefore added a detailed proof in the supplement.

Comparison PG/mPG/PGAS/mPGAS: PGAS/mPGAS are now implemented in Birch. We have chosen to focus on

the performance improvement offered by marginalization. On the toy model we observe a clear improvement from

marginalizing both for PG and PGAS, but also from using PGAS/mPGAS compared to using PG/mPG. However, for

the VBD model we observe a clear improvement from marginalizing, but using PGAS/mPGAS instead of PG/mPG

gives no clear improvement. Results supporting these claims have been added to the supplement.

Computational cost: There are indeed some extra computations for the marginalized methods, but the overhead is quite

small. For the toy model, with N=500, using the tic-toc timer in MATLAB we get: PG 1231.5 s mPG 1430.7 s PGAS

1260.7 s mPGAS 1566.1 s. Note that the code has not been optimized.

Line 45-56, misleading discussion: This discussion is mainly intended as a pedagogical motivation for why marginal-

ization is useful. PG is the standard approach in many cases, but is limited by the "ideal" Gibbs it approximates. What

we propose to do instead is, like you point out, to approximate the "ideal" collapsed Gibbs sampler using marginalized

versions of PG/PGAS. Note that "ideal" here does not mean optimal, but refers to the hypothetical non-particle version.

We have clarified this in the paper, and to avoid confusion we have changed the word "ideal" to "hypothetical".

**Reviewer #2**

Using backward sampling?  Yes that is possible, however, [1] argue that ancestor sampling is more suitable for

non-Markovian models and we have therefore chosen to focus on PGAS.

Path degeneracy: Indeed, ancestor sampling helps to reduce the effect of path degeneracy. Furthermore, the MCMC

nature of mPG/mPGAS means that we can "revisit" and update states at early time steps, which is not possible in purely

"online" methods, which also mitigates the effect of path degeneracy.

**Reviewer #3**

Extend the evaluation with more large models and on more datasets: The VBD implementation has been extended to

daily data (instead of weekly) and we have updated relevant figures in the paper with these. All conclusions are the

same as before. Regarding running times, see the reply to Reviewer #1.

Clearer definitions: (1) Definitions of h,s, and A are the same as for the exponential family (defined in the supplement).

For clairity we have added definitions of all variables in the main text and a more detailed explanation of the restricted

exponential family is now in the supplement. (2) $\tilde{w}^i$ is the weight of each possible ancestor trajectory, and is used in the

resampling step to assign a new ancestor path to the reference trajectory. A detailed description and motivation can be

found in [1]. We have clarified this in the paper.

Figure 1: Ideally we would like iid samples from the posterior distribution, in terms of the ACF of the samples it should

be zero everywhere except for lag 0. Figure 1 illustrate that mPGAS yields samples with a lower autocorrelation (closer

to iid) than what is attainable with standard PGAS, even for a low number of particles ($N = 50$).

Path degeneracy of PGAS (line 81): We viewed path degeneracy as the issue that the mixing rate goes to zero as $T$

becomes large (for fixed $N$), which is typically not the case for PGAS. However, the mixing of PGAS is indeed affected

by the mixing properties of the model. Typically, it decreases to a *non-zero constant* as $T$ becomes large. To avoid

confusion we have changed the wording from "unaffected" to "more robust".

**References**

[1] F. Lindsten, M. I. Jordan, and T. B. Schön. Particle Gibbs with ancestor sampling. *JMLR*, 15:2145–2184, 2014

[2] D. A. van Dyk and T. Park. Partially collapsed gibbs samplers. *JASA*, 103(482):790–796, 2008


[Meta-Review · NeurIPS 2019]

This paper makes a solid contribution to improving inference in certain state space models that a used extensively in practice, particularly when implementing such models in a probabilistic programming language.